# Light Quality-Dependent Regulation of Non-Photochemical Quenching in Tomato Plants

**DOI:** 10.3390/biology10080721

**Published:** 2021-07-28

**Authors:** Magdalena Trojak, Ernest Skowron

**Affiliations:** 1Department of Medical Biology, Jan Kochanowski University, Uniwersytecka 7, 25-406 Kielce, Poland; 2Department of Environmental Biology, Jan Kochanowski University, Uniwersytecka 7, 25-406 Kielce, Poland; ernest.skowron@ujk.edu.pl

**Keywords:** antioxidant enzyme, chlorophyll fluorescence quenching, indoor farming, light quality, LED, non-photochemical quenching

## Abstract

**Simple Summary:**

Photosynthetic organisms, such as land plants, evolved to utilize available light and to use its energy to assimilate carbon dioxide and produce carbohydrates. However, the light intensity often exceeds the ability of plants to successfully utilize absorbed energy, thus inducing stress, manifested by an increased radical concentration inside plant cells and disruption of the inner structures, and consequently decreased plant yield. Plants solve this problem by using a mechanism termed non-photochemical quenching, by which they can dissipate the energy not used in photosynthesis. Modern agriculture, however, also involves indoor plant farming. For indoor plant farming LED-based lighting systems, with non-saturating light intensities, are suitable based on their restricted energy consumption. However, the composition of applied light should first be optimized to maximize its utilization. Our study examined the influence of monochromatic LEDs (red, green, and blue) on the photoprotective and photosynthetic properties of tomato plants. We indicate that monochromatic green light could be considered an important component of lighting systems to alleviate energy dissipation, while blue light enhances photosynthetic efficiency. Our study not only proves the crucial importance of spectrum optimization but also provides evidence that different light wavelengths modify photosynthetic and photoprotective properties.

**Abstract:**

Photosynthetic pigments of plants capture light as a source of energy for photosynthesis. However, the amount of energy absorbed often exceeds its utilization, thus causing damage to the photosynthetic apparatus. Plants possess several mechanisms to minimize such risks, including non-photochemical quenching (NPQ), which allows them to dissipate excess excitation energy in the form of harmless heat. However, under non-stressful conditions in indoor farming, it would be favorable to restrict the NPQ activity and increase plant photosynthetic performance by optimizing the light spectrum. Towards this goal, we investigated the dynamics of NPQ, photosynthetic properties, and antioxidant activity in the leaves of tomato plants grown under different light qualities: monochromatic red (R), green (G), or blue (B) light (L) at 80 µmol m^−2^ s^−1^ and R:G:B = 1:1:1 (referred to as the white light (WL)) at 120 µmol m^−2^ s^−1^. The results confirm that monochromatic BL increased the quantum efficiency of PSII and photosynthetic pigments accumulation. The RL and BL treatments enhanced the NPQ amplitude and showed negative effects on antioxidant enzyme activity. In contrast, plants grown solely under GL or WL presented a lower amplitude of NPQ due to the reduced accumulation of NPQ-related proteins, photosystem II (PSII) subunit S (PsbS), PROTON GRADIENT REGULATION-LIKE1 (PGRL1), cytochrome *b*_6_*f* subunit *f* (cyt*f*) and violaxanthin de-epoxidase (VDE). Additionally, we noticed that plants grown under GL or RL presented an increased rate of lipid peroxidation. Overall, our results indicate the potential role of GL in lowering the NPQ amplitude, while the role of BL in the RGB spectrum is to ensure photosynthetic performance and photoprotective properties.

## 1. Introduction

The initial step of photosynthesis involves the absorption of light. However, when the amount of light absorbed exceeds the capacity of its photosynthetic utilization, this can lead to the formation of reactive oxygen species (ROS) and damage to the photosynthetic apparatus [1,2]. To reduce the risk of potential damage plants evolved protective mechanisms, including ways to minimize light absorption, detoxify ROS and dissipate excess light energy as heat [3]. This thermal dissipation process, called non-photochemical quenching (NPQ), acts as a safety valve. NPQ includes components with different mechanisms and characteristic times—energy-dependent (qE), zeaxanthin (Z)-dependent (qZ), and photoinhibitory (qI)—as well as components related to state transition (qT), chloroplast movement (qM) [4], and the newly termed plastid lipocalin-dependent photoprotective antenna quenching (qH) [3]. Extensive research, devoted to understanding the mechanism of NPQ components might be justified, as NPQ has been considered to be a prime target to increase light use efficiency [5]. 

First, energy-dependent quenching, qE, is turned on rapidly by an increase in thylakoid lumen proton concentration, followed by protonation of the thylakoid membrane protein, photosystem II (PSII) subunit S (PsbS) [6], and reversible conversion of xanthophyll pigments. qE follows the variations of the transmembrane proton gradient (ΔpH), thus its induction and relaxation typically develop within a few minutes (1–3 min) [7,8]. For qE, it is proposed that PsbS promotes energy quenching as it triggers the undocking of light-harvesting complex II (LHCII) from the PSII core [9], thus promoting antenna oligomerization. Protonation of PsbS is initiated by linear electron transport (LET) and cyclic electron transport (CET) coupled with the proton pumping activity of cytochrome *b*_6_*f* (cyt*b_6_f*) [10].

Second, qZ, another NPQ component, is detectable within 10–15 min and is related to violaxanthin (V) de-epoxidation [9] but not to PsbS [8]. Thus, the qZ component is strictly related to the xanthophyll cycle activity [11]. The drop in lumen pH activates violaxanthin de-epoxidase (VDE) and shifts the xanthophyll balance from V which acts as a light-harvesting pigment, toward Z favoring dissipation of the excitation energy through heat [12]. However, while qE relaxation is fully activated by the collapse of ΔpH, qZ relaxation is prolonged, and qZ NPQ appears to remain active even in the absence of a low lumen pH [13]. 

Third, the photoinhibitory component of NPQ, qI, is slowly reversible (within hours) and comprises processes related to either PSII core inactivation characterized by reduced maximal photochemical efficiency (Fv/Fm) [14] or slowly relaxing quenching, recently termed qH [3]. The qT component, which indicates state transitions, is triggered by low light conditions to balance the energy distribution of light energy between PSII and photosystem I (PSI) [15].

Moreover, previous research also proved, that the antioxidant capacity of plant cells interferes with NPQ activity [1] and both are related to spectrum quality [16,17]. The relationship between antioxidants and NPQ might be explained by considering the VDE enzyme, the domain of which has been suggested to bind the main substrate V and the co-substrate ascorbic acid (AsA) as a reductant [11,18]. This is because, AsA also has a photoprotective function as a cofactor for antioxidant enzymes, such as ascorbate peroxidase (APX) [1], in scavenging ROS. Photoreduction of oxygen by PSI to a superoxide anion radical is followed by its dismutation to hydrogen peroxide (H_2_O_2_) and oxygen by superoxide dismutase (SOD). Then H_2_O_2_ is reduced by APX to water [1]. Like APX, catalase (CAT) also detoxifies H_2_O_2_; however, CAT mainly occurs in peroxisomes and does not require a reductant (AsA) to catalyze dismutation reactions [19]. Consequently, increased APX activity is an important factor in reducing the ascorbate availability for VDE and in decreasing the amplitude of NPQ. On the other hand, VDE activity can be artificially restricted by the well-known reductant agent dithiothreitol (DTT) [20] to study NPQ dynamics. The mechanism of DTT is likely to be related to its reducing action on disulfide bonds in the VDE molecule [20]. However, DTT does not interfere with the development of the trans-thylakoidal ΔpH [8] regulating qE. Thus, feeding leaves DTT and AsA, which affect VDE activity in opposite ways, could be applied to distinct NPQ components for a better understanding of the influence of light quality on NPQ characteristics and amplitude.

Overall, our current understanding of NPQ activity suggests, that when operating in field conditions it determines canopy photosynthesis, biomass, and yield by preventing photoinhibition in response to rapidly changing environmental conditions [21]. However, under non-saturating light conditions in indoor farming systems (e.g., LED-based), more of the absorbed energy can be used for photochemistry when NPQ is not highly induced [22]. Thus, it would be favorable to restrict NPQ activity and to increase the speed of its recovery during relaxation [5], as its overly-protective persistence lowers the quantum yield of photosynthesis [21]. The effect of different light quality on NPQ amplitude has not been thoroughly investigated [5], even though NPQ activation is linked to the absorption of mostly blue (B) and red (R) light by photosynthetic pigments [22]. For this purpose, the RB spectrum supplemented with green (G) light could be considered to ensure efficient photosynthesis, avoiding needless NPQ activation.

In the present study, we investigated the influence of the spectral quality of light on the dynamics of non-photochemical quenching, photosynthetic properties, and the absorbed energy partitioning in PSII in the presence of DTT and AsA in the leaves of tomato plants. For this purpose, we examined the kinetics of chlorophyll fluorescence induction and dark-relaxation and, in particular, proposed an original model of the fast induction and relaxation curves. We also analyzed the antioxidant activity of APX, SOD, and CAT enzymes, the photosynthetic pigment content, and the abundance of proteins related to the xanthophyll cycle (PsbS and VDE) or linear (cytochrome *b*_6_*f* subunit *f*; cyt*f*) and cyclic electron transport (PROTON GRADIENT REGULATION-LIKE1; PGRL1) building up the proton gradient across the thylakoid membrane. We hypothesized the following: (1) different types of monochromatic light have different impacts on the dynamics of non-photochemical quenching and that response is related to (2) different patterns of absorbed energy partitioning in PSII, (3) altered accumulation of photosynthetic pigments and NPQ-related proteins and (4) modified antioxidant capacity. The use of monochromatic R, G, and B light along with RGB (WL) lighting represents an opportunity to increase/decrease the proportions of desired wavebands in the mixed spectrum, thus increasing plant performance.

## 2. Materials and Methods

### 2.1. Plant Material and Growth Conditions

Tomato (*Solanum lycopersicum* L. cv. Malinowy Ozarowski) seeds, treated with antifungal powder (T75 DS/WS) were germinated in Petri dishes on sterile filter papers soaked in Milli-Q-water at 26 °C. For analysis, a tomato cultivar with reduced leaf dissection (potato leaf phenotype) [23] was chosen to maximize light absorption in the upper part of the canopy. The seedlings were transplanted to P9 containers (9 × 9 × 10 cm) and filled with the substrate (white and black peat, perlite and N:P:K = 9:5:10; pH 6.0–6.5), divided into groups, and transferred to four growth chambers, with non-reflective black separators to eliminate light contamination. The plants were grown for 14 consecutive days under Px256 PxCrop LED lamps (PXM, Podleze, Poland) delivering 120 µmol m^–2^ s^–1^ of the RGB spectrum (R:G:B = 1:1:1; referred to as white light (WL)) or 80 µmol m^–2^ s^–1^ of monochromatic R, G or B light (RL, GL or BL, respectively). LED characteristics were as follows: red LEDs: peak wavelength 671 nm, peak broadness at half peak height 25 nm (656–681 nm); green LEDs: 524, 40, 505–545 nm; and blue LEDs: 438, 20, 428–448 nm. WL treatment was used as the control group. Light composition and photosynthetic photon flux density (PPFD) were monitored daily by a calibrated spectroradiometer GL SPECTIS 5.0 Touch (GL Optic Lichtmesstechnik GmbH, Weilheim/Teck, Germany). The readings were averaged for six locations at the level of the apical bud and maintained by adjusting the distance between the light sources and the plant’s canopy. 

The containers with tomato plants cultivated under the same light treatment were turned twice a day. To avoid canopy shading and overlapping five plants per square meter of the illuminated area were cultivated. The photoperiod was 16/8 h (day/night; day 6:00 a.m.–10:00 p.m.), the average temperature was maintained at 25/22 °C (day/night) and relative air humidity was kept at 50–60%. The plants were watered with tap water when necessary and fertilized once a week with 1% (*w/v*) tomato fertilizer (N:P:K = 9:9:27; Substral Scotts, Warszawa, Poland). The second leaf from above of plants 14-days-after-transplanting (DAT) was used for subsequent analyses. All analyses were conducted between 8:00 a.m. and 12:00 p.m. Twenty tomato plants (two repetitions with ten plants per light treatment) were grown with each kind of light treatment.

### 2.2. Pre-Illumination of Ascorbic Acid- and Dithiothreitol-Infiltrated Leaf Samples

Leaf discs with a diameter of 10 mm were cut out (avoiding leaf veins) from the second leaf from above of plants 14-DAT in each group, transferred to Petri dishes, and incubated in a solution of 10 mM ascorbic acid (AsA, VDE enzyme activator; Sigma-Aldrich, St. Louis, MO, USA) [1] or 5 mM dithiothreitol (DTT, VDE enzyme inhibitor; Sigma-Aldrich) [24] or in distilled water (control, C) for 30 min in the dark. All solutions were buffered with 10 mM piperazine-1,4-bis(2-ethanesulphonic acid) (PIPES; Sigma-Aldrich) [25]. To avoid light contamination discs were cut out directly under the lighting conditions of the chambers. After infiltration, leaf discs were pre-illuminated for 30 min with RGB light (R(627 nm): G(530 nm): B(447 nm) = 1:1:1) at a light intensity of approximately 500 µmol m^−2^ s^−1^ ± 5 µmol m^−2^ s^−1^ (LED Light Source SL-3500 lamp, Photon Systems Instruments, Drasov, Czech Republic). Light intensity applied for pre-illumination was chosen based on previous analyses [26,27] to allow for the most pronounced acceleration of NPQ formation and to avoid any adverse effect on PSII activity at higher light intensities. Subsequently, leaves were re-darkened for 15 min to allow relaxation of the transthylakoid pH gradient without substantial reconversion of Z back to V.

### 2.3. Measurement of Chlorophyll Fluorescence (ChF) Induction Kinetics

ChlF was measured at 25 °C using a pulse amplitude-modulated (PAM) fluorometer (Maxi IMAGING PAM M-Series, Walz, Effeltrich, Germany) on the adaxial side of the leaf discs. Leaf discs were positioned on wet filter paper in a self-built cuvette and continuously supplied with moistened air throughout the experiment. To avoid spatial and temporal heterogeneity of chemical infiltration, four circle-shaped areas of interest (AOIs) were selected and averaged for each replicate. The minimal (dark) fluorescence level (Fo) was measured using measuring modulated blue light (450 nm), which was sufficiently low (0.01 µmol m^−2^ s^−1^) that it would not induce any significant variable fluorescence. The maximal fluorescence level (Fm) with all PSII reaction centers closed was determined by a 0.8 s saturating blue light pulse (SP = 450 nm) at 5000 µmol m^−2^ s^−1^ in dark-adapted samples. The maximum PSII photochemical efficiency (Fv/Fm) was derived from that (Fv/Fm = (Fm − Fo)/Fm). Then, for quenching analysis, leaf samples were illuminated for 20 min at 185 µmol m^–2^ s^–1^ of blue actinic light (AL = 450 nm) (induction), followed by 20 min of dark incubation (relaxation). AL intensity during the measurement was chosen to be just sufficient to provide stable NPQ amplitude over the induction phase [26]. To determine the NPQ induction, SPs (5000 µmol m^−2^ s^−1^, duration 0.8 s) were applied every 10 s apart during the first 50 s of AL illumination (fast induction), followed by five flashes given every 240 s (slow induction). Subsequently, the AL was turned off and the dark relaxation of NPQ was determined by SPs spaced 10 s during the first 50 s of dark incubation (fast relaxation) followed by five flashes given every 240 s (middle and slow relaxation). The saturating light flashes were time-separated to minimize their contribution to NPQ formation and relaxation.

Fluorescence yields obtained during the analysis were combined to calculate the Stern–Volmer NPQ = (Fm − Fm′)/Fm′ (where Fm′ is the maximal level of chlorophyll fluorescence in light) and the complementary effective quantum yield of PSII photochemistry ΦPSII = (Fm′ − F)/Fm′ (where F is fluorescence yield in conjunction with an applied saturation pulse) [28], the non-regulated energy dissipation ΦNO = 1/(NPQ + 1 + qL(Fm/Fo − 1)) (where qL is the coefficient of photochemical quenching based on the *lake model* of PSII antenna pigment organization) and regulated energy dissipation ΦNPQ = 1 − ΦPSII − ΦNO [8,29]. The assessed NPQ value exceeded unity; thus, its value is presented as NPQ/4 for better correspondence with ΦNPQ. Each measurement comprised six replicates.

### 2.4. Antioxidant Enzyme Activity Assay and MDA Measurements

Fresh leaf samples, collected before pre-illumination, were immediately frozen in liquid nitrogen and ground to a fine powder using a chilled mortar and pestle. Soluble proteins were extracted by homogenizing 100 mg of leaf powder in 1 mL of ice-cold 50 mM phosphate buffer (pH 7.8 for SOD and 7.0 for APX/CAT assay) containing 1 mM EDTA (pH 8.0) and 5% (*w*/*v*) polyvinylpolypyrrolidone (PVPP) and 1 mM ascorbate (for APX assay). Insoluble material was removed by centrifugation at 10,000× *g* for 30 min at 4 °C. The collected supernatant, referred to as the “extract”, was used to determine antioxidant enzyme activity. 

For ascorbate peroxidase (APX; EC 1.11.1.11) determination, previously described assays [30,31] were performed. The reaction mixture, containing 50 mM phosphate buffer (pH 7.0), 1 mM sodium ascorbate, and 50 μL of the extract, was first equilibrated for 3 min. The reaction was started by the addition of 0.5 mM H_2_O_2_ and monitored by the decrease in absorbance at 290 nm due to ascorbate oxidation every 30 s for 3 min at 25 °C (extinction coefficient of 2.8 mM^−1^ cm^−1^). The assay was performed in four replicates for each treatment.

The activity of superoxide dismutase (SOD; EC 1.15.1.1) was determined using the previous assay [32]. For the assay, 40 µL of each extract was mixed with 100 μL of 0.75 mM nitroblue tetrazolium (NBT), 2 µL of 0.5 M EDTA (pH 8.0), and 20 μL of 0.1 mM riboflavin. Due to the photosensitivity of the solution, the procedure was carried out under low light. Samples containing the reaction solution were irradiated under a set of fluorescent light tubes of 40 µmol m^−2^ s^−1^ for 10 min to start the reaction. The absorbance of the irradiated and non-irradiated samples was determined at 560 nm. One unit of enzyme activity (U) was taken as the amount of enzyme that reduced the absorbance reading to 50% compared with tubes that lacked the enzyme [33]. The assay was performed in four replicates for each treatment.

The activity of catalase (CAT; EC 1.11.1.6) was determined as described elsewhere [34]. In brief, 20 μL enzyme extract was added to 405 μL 50 mM potassium phosphate buffer (pH 7.0) and 250 μL water. After 5 min of pre-incubation at 26 °C, 750 μL of 10 mM H_2_O_2_ was added to start the reaction. The decomposition of H_2_O_2_ was followed directly by decreased absorbance at 240 nm. Enzyme activity was computed by calculating the decomposed amount of H_2_O_2_, every 20 s for 3 min at 26 °C. The calculations used an absorbance coefficient of 43.6 M^−1^ cm^−1^ at 240 nm [35]. The assay was performed in four replicates for each treatment.

The level of oxidative damage to membranes was estimated indirectly by assessing the by-products of lipid peroxidation reacted with thiobarbituric acid (TBA), including malondialdehyde (MDA) content. The assay was in accordance with a previous procedure [32]. First, 100 mg of liquid nitrogen-ground leaf powder was homogenized in 1 mL of 10% (*w*/*v*) TCA and centrifuged at 10,000× *g* for 15 min. Then, 1 mL of supernatant was mixed with 1 mL of 0.6% (*w*/*v*) TBA, heated at 95 °C for 30 min, and then quickly cooled down on ice to room temperature. After centrifugation at 10,000× *g* for 10 min, the absorbance (A) of the 10-fold diluted supernatants was measured at 532 nm and values corresponding to non-specific absorption at 450 nm and a correction factor for non-specific turbidity at 600 nm. MDA concentration determined on a fresh weight basis was calculated as described elsewhere [36] with the following Formula (1): MDA (µmol g^−1^ FW) = 6.45 × (A_532_ − A_600_) − 0.56 × A_450_(1)

The assay was performed in four replicates for each treatment.

### 2.5. Pigment Determination

The concentrations of chlorophyll *a* and *b* (Chl *a*, *b*) and total carotenoids were measured spectrophotometrically with a Spectronic Helios Gamma UV-Vis spectrophotometer (Thermo Fisher, Waltham, MA, USA) after being dissolved in dimethyl sulfoxide (DMSO). Pigments were extracted from leaf discs (3 mm in diameter, approximately 20 mg of tissue) in 1.5 mL DMSO. Samples, kept in dim light, were vortexed for 1 min, then capped and incubated for 3 h at 65 °C with inversion every 10 min to improve extraction. Then the sample mixture was centrifuged at 10,000× *g* for 15 min, and the supernatant was carefully collected without disturbing the plant tissue, transferred to a new tube, and mixed again for 15 s. An aliquot (1 mL) of the uppermost supernatant layer was used for pigment determination at 480, 649, and 665 nm, according to an optimized method described elsewhere [37]. The assay was performed in ten replicates for each treatment.

### 2.6. Western Blot Analysis

Leaf-soluble proteins were extracted with a Plant Total Protein Extraction Kit (Sigma-Aldrich) according to the manufacturer’s instructions. In brief, 200 mg of the liquid nitrogen-ground leaf powder, protected from proteolysis by a protease inhibitor cocktail, was washed with a methanol working solution and acetone. A purified tissue pellet was used for total protein extraction with a chaotropic protein reagent. The protein content was estimated using Coomassie reagent (Thermo Fisher Scientific) and bovine serum albumin as a standard [38]. 

Then, optimized amounts of extracted proteins were loaded onto precast 4–20% gradient TGX polyacrylamide gels (Bio-Rad, Hercules, CA, USA) and run with a constant voltage of 200 V for 20 min. Separated proteins were transferred to nitrocellulose membranes (0.45 or 0.2 µm pore size; Bio-Rad) by semi-dry electroblotting (1.5 mA per cm^2^, 20 min). Air-dried blots were blocked with 5% non-fat dry milk blocking reagent (1 h, 25 °C) (Bio-Rad) and incubated with primary antibodies against PsbS (AS09 533; 1:1000), VDE (AS15 3091; 1:1000), PGRL1 (AS10 725/AS19 4311; 1:1000), cyt*f* (AS08 306; 1:5000) and ATPB (beta subunit of ATP synthase; AS05 085; 1:5000; loading control) (Agrisera, Vännäs, Sweden) overnight at 4 °C. Next membranes were washed in Tween-TBS buffer (TTBS; 0.05% Tween 20, 20 mM Tris, 500 mM NaCl) and incubated with horseradish peroxidase-conjugated secondary antibody (AS09 602; 1:5000–1:10,000) for 1 h at 25 °C with agitation. The blots were washed again in TTBS and developed for 5–10 min with a colorimetric detection reagent using a Pierce™ DAB Substrate Kit (3,3′-diaminobenzidine tetrahydrochloride; Thermo Fisher Scientific). Quantification of the protein bands of the Western Blot (WB) membranes visualized with DAB was done using densitometric analysis (ImageJ v.1.49, National Institutes of Health, Bethesda, MD, USA) [39]. The samples were analyzed three times. The relative amount of proteins was calculated using the maximum value of protein noticed in WL plants.

### 2.7. Model for Fitting of Experimental Data of Fast qE NPQ

Curve fitting of NPQ induction and relaxation was performed using Origin version 2021b (OriginLab Corporation, Northampton, MA, USA). The model for fitting experimental data of fast (50 s) qE NPQ induction/relaxation in AsA-, DTT- or water-treated samples, was applied as specified for each case and reported with the adjusted R^2^ (R_a_^2^) value to determine the goodness of data fitting (Table 1 and Table 2). We applied the following:Logistic regression equation for fitting data (*x*) of fast qE NPQ induction:
(2)$$y=\frac{A_{1}-A_{2}}{1+{\left(\frac{x}{x_{0}}\right)}^{\alpha }}+A_{2}$$
where *A*_1_ is the minimum asymptote, *A*_2_ is the maximum asymptote, *x*_0_ is the value of the inflection point and *α* is the slope of the logistic growth rate (steepness of the curve).

Polynomial cubic regression equation for fitting data (*x*) of fast qE NPQ relaxation:

(3)$$y=A+Bx+Cx^{2}+Dx^{3}$$

where *A* is the offset and *B*, *C* and *D* are coefficients.

### 2.8. Statistical Analysis

Statistical analysis was performed using Statistica 13.3 software (StatSoft Inc., Oklahoma, OK, USA). The normal distribution of variables was verified using the Shapiro–Wilk test and the equality of variances was evaluated using Levene’s test. One-way ANOVA and post-hoc Tukey’s HSD tests were employed to analyze the differences between the investigated groups. The data are presented as mean with standard deviation (±SD). Statistical significance was determined at the 0.05 level (*p* = 0.05).

## 3. Results

### 3.1. NPQ Formation in the Presence of AsA and DTT

The induction of NPQ during 20 min of illumination at a light intensity of 185 µmol m^−2^ s^−1^ was measured in tomato plants grown under different types of light quality: monochromatic R (RL), G (GL), or B (BL) light at 80 µmol m^−2^ s^−1^, or combined RGB spectrum (WL) at 120 µmol m^−2^ s^−1^. To overcome possible limitations of NPQ formation by the rate of Z synthesis, especially in plants grown under monochromatic light, NPQ dynamics were investigated after pre-illumination of leaf samples for 30 min with RGB light at 500 µmol m^−2^ s^−1^. Before the onset of pre-illumination, leaves were infiltrated with 10 mM AsA or 5 mM DTT, or incubated in distilled water (control, C). DTT is an inhibitor of VDE that promotes NPQ induction, whereas AsA is a cofactor for the VDE enzyme. This approach allows us to examine the ability of plants grown under different light quality in terms of the subsequent Z formation during pre-illumination and evaluate the contribution of the NPQ components to the overall non-photochemical activity within re-darkened samples.

In water-incubated samples, a stable maximum NPQ value was reached after about 10 min (600 s) of AL illumination (Figure 1). However, analyzed NPQ kinetics revealed significant differences in maximal amplitude between groups. The highest NPQ level was observed in BL and RL plants, reaching 1.0 and about 0.84, respectively, whereas, in WL and GL plants, the maximum level stabilized at 0.55 and 0.34, respectively, after 600 s. Moreover, an analysis of the kinetics of NPQ induction revealed the existence of two main phases, rapid (development time about 50 s) and slower (development time about 600 s). 

The results of the rapid phase of NPQ induction, assigned to qE, are described below. In the case of the slower phase assigned to qZ, its amplitude contributed to about 20% of the total NPQ value (19, 20, 24, and 16% in WL, RL, GL, and BL samples, respectively). DTT infiltration, however, significantly reduced the NPQ amplitude. The fraction of lost NPQ might be attributed to the Z synthesized with VDE, as the amplitude of quenching in DTT-treated samples did not differ significantly between 50 and 600 s of AL illumination. In contrast, AsA infiltration significantly increased the maximal NPQ amplitude, but only in the GL leaves (Figure 1).

Similarly, the relaxation of NPQ presented distinct phases: rapid, which contributed to the first 50 s of relaxation (1250–1300 s), middle (4 min, 1300–1540 s), and slow (last 16 min, 1540–2500 s) (Figure 1). The results of rapid and middle NPQ relaxation, related mostly to pH collapse, are described below. In the case of the slow phase of NPQ relaxation, it was found to collapse almost completely after 20 min of dark incubation, indicating that the reaction of Z epoxidation back to V was not interrupted by the applied chemicals.

### 3.2. Kinetics of Rapid Phase of NPQ Induction

The dominant rapid phase (qE) of water-incubated samples reached values of 0.45, 0.67, 0.26, and 0.82 for WL, RL, GL, and BL, respectively, after about 50 s of AL illumination. At the same time, DTT infiltration was unable to completely suppress the NPQ rise, however, the leaf samples presented a significantly reduced qE amplitude, reaching 0.15, 0.25, 0.17, and 0.29 for WL, RL, GL, and BL, respectively (Figure 2). In general, the initial slope of the NPQ fast phase formation in water-treated samples was much steeper in the BL and RL leaves (*α*_BL_= *α*_RL_ = 3.89) than that in WL leaves (*α*_WL_ = 2.79) (Table 1, Figure 2), reaching a near-maximal value after 30 s of AL illumination.

On the contrary, in the GL samples, the qE rise was distinctly slowed down (*α*_GL_ = 2.31). Despite the higher PPFD as well as the red and blue light included in the spectrum, the WL samples also presented a slower qE rise than RL or BL. Moreover, in the BL and RL samples, DTT feeding reduced not only the qE amplitude but also the steepness of the curve (to *α =* 2.35 in both light conditions). In contrast, DTT-fed WL leaves presented only slightly decreased *α*, while GL showed an increased dynamic of qE rise, despite its lower maximal amplitude (Table 1). On the other hand, the AsA infiltration, which was expected to induce Z formation during the pre-illumination phase and thus to accelerate qE NPQ formation, visibly increased the qE amplitude only in GL samples, whereas the WL- and BL-grown leaves presented diminished qE values after the first 50 s. However, in AsA-fed GL samples, qE quickly reached a plateau (the inflection point, *x*_0_, occurred after the first 6 s of induction), compared to WL (Table 1).

### 3.3. Kinetics of Rapid and Middle Phases of NPQ Relaxation

An analysis of the kinetics of the fast NPQ relaxation demonstrated that it was rather slow, especially in the GL group (Table 2, Figure 3), when compared to the fast induction of NPQ. During the first 50 s, the amplitude of NPQ relaxed about 25, 27, 18, and 39% in control conditions for WL, RL, GL, and BL samples, respectively. After the next 4 min of dark incubation, the NPQ amplitude further decreased by about 72, 79, 71, and 82% for WL, RL, GL, and BL samples, respectively. DTT-infiltrated samples presented similar amplitudes of NPQ relaxation during the fast phase (17, 39, 27, and 36% for WL, RL, GL, and BL samples, respectively), whereas the rate of the middle phase of relaxation was significantly restricted (33, 28, 43 and 42% for WL, RL, GL and BL samples, respectively). In the case of AsA-infiltrated leaves, the relaxation dynamics of rapid and middle phase were similar to those noted for control (water) samples, except for BL, which presented a significantly reduced amplitude (21% compared to 39%) of the fast NPQ relaxation. Thus, we speculate that AsA stimulates the formation of Z during AL illumination; therefore, the relaxation of NPQ activated by the collapse of ΔpH is prolonged due to higher Z accumulation. Overall, the results document that the rapid and middle phases of NPQ relaxation can be predominantly assigned to the qE component and represent pH-regulated and PsbS-dependent processes [27]. Thus, the slower rate of NPQ relaxation within DTT-fed leaves might be attributed to restricted proton consumption as measured within Fv/Fm (Figure 4) and ΦPSII (Figure 5). However, the rate of middle phase relaxation also seems to be related to the pool of Z as documented for AsA-infiltrated BL leaves. 

### 3.4. Maximum Quantum Yield and Photosynthetic Energy Partitioning in PSII

For unstressed leaves, the value of Fv/Fm photosynthesis is highly consistent, with a value of about 0.83, and correlates to the maximum quantum yield of photosynthesis, while the existence of any type of “stress” lowers Fv/Fm [40]. An analysis of Fv/Fm demonstrates that light quality influenced the activity of PSII (Figure 4). The highest Fv/Fm value of water-incubated samples was observed in plants grown under monochromatic blue (Fv/Fm = 0.751) and the lowest under green (Fv/Fm = 0.629) and red (Fv/Fm = 0.634) light. At the same time, white light-grown leaves presented a higher Fv/Fm value (0.711) than GL or RL, revealing the positive influence on PSII vitality of including B light in the spectrum. DTT infiltration did not cause serious PSII damage, with decreased Fv/Fm by about 4, 5, and 3% for WL, GL, and BL, respectively. On the contrary, AsA-infiltrated leaves presented increased Fv/Fm in GL and BL by about 13 and 5%, respectively. It should be noted, however, that the observed Fv/Fm values, in all treatments, were lower than expected indicating stressful conditions. A likely explanation is that leaf samples of plants grown under low light conditions were subsequently exposed to relatively high-intensity light to accelerate NPQ formation.

The partitioning of excitation energy was then analyzed to assess the contribution of the mechanism of regulated energy quenching (ΦNPQ) (Figure 6) and to distinguish it from that of non-regulated quenching (ΦNO) (Figure 7) and photochemistry yield (ΦPSII) (Figure 5). The amplitude of the quantum yield of the regulated energy dissipation expressed as ΦNPQ, was similar to that measured with NPQ. However, in water-incubated samples, the ΦNPQ rose to 0.54, 0.77, 0.57, and 0.8 after 600 s of AL illumination in WL, RL, GL, and BL samples, respectively (Figure 6). We also noticed that the amplitude of ΦNPQ in WL reached a maximum after about 50 s and then partially relaxed to a lower value within 200 s. Such relaxation, which might be attributed to the appearance of transient energy-dependent quenching (qE_TR_), was not observed in plants grown under monochromatic light, presumably due to the lower PPFD applied for their cultivation. Although DTT-fed leaves in all light treatments presented a suppressed ΦNPQ rise compared to control, the difference was less pronounced than in the NPQ analysis. Similarly, the positive effect of AsA feeding on ΦNPQ amplitude in GL samples was also mitigated compared to NPQ analysis.

As the fraction of absorbed light energy added up to unity the reduced ΦNPQ amplitude was related to concomitant ΦNO rise (Figure 7), reflecting the contribution of absorbed excitation energy used by neither photochemistry nor protective regulatory mechanisms. Thus, on the same timescale, the ΦNO values accounted for 25, 23, 43, and 20% of energy quenching for WL, RL, GL, and BL samples, respectively (Figure 7). Consequently, during the induction phase in all plants grown under monochromatic light, the kinetics of ΦPSII showed rapid inhibition (Figure 5) due to the saturating intensity of AL. In the case of WL plants, grown in higher PPFD, ΦPSII accounted for about 20% of light energy utilization. However, during the first 600 s of the relaxation phase, the ΦPSII values were restored to 0.67, 0.57, 0.56, and 0.70 for WL, RL, GL, and BL, respectively. The lower rate of ΦPSII with a concomitant higher ΦNO value during recovery in GL and RL leaves was related to the already decreased Fv/Fm, noted prior to the analysis of NPQ dynamics.

### 3.5. Photosynthetic Pigment Accumulation

Decreased concentrations of chlorophyll *a* and *b* (Chl *a* and *b*) were detected in leaves of the GL group, but the former only when compared to BL plants (Table 3). Consequently, cumulative chlorophyll concentration was lower in GL leaves than in WL and BL leaves by 13 and 15%, respectively. Like GL plants, RL also presented a tendency for decreased content of both chlorophylls, however, there was no statistically significant difference compared to WL. Despite the differences in chlorophyll accumulation, spectrum quality did not affect the Chl *a*/*b* ratio. However, the content of carotenoids, which were estimated as a pool, was also found to be the lowest in GL plants (Table 3).

### 3.6. Activity of Antioxidant Enzymes APX, SOD, and CAT; MDA Accumulation

The activity levels of antioxidant enzymes in tomato leaves are presented in Figure 8. APX activity in the GL treatment was about 25% higher than that in the WL treatment. However, in other groups, APX activity declined by approximately 38 and 15% under RL and BL, respectively (Figure 8a). Monochromatic green light also showed a positive effect on SOD in relation to other treatments, presenting 39% higher activity compared to WL plants (Figure 8b). At the same time, there was no significant difference in SOD activity between RL and BL treatments; however, SOD activity was still about 17 and 24% lower under RL and BL, respectively, than under WL. Thus, an increase in SOD activity in WL indicates that GL supplementation led to a concomitant stimulation of antioxidant properties. At the same time, no statistically significant differences were found in CAT activity among light treatments (Figure 8c). However, we noticed that light quality affected MDA accumulation, related to the rate of lipid peroxidation in the leaf tissue. Consequently, the highest MDA levels were noticed in leaves developed under GL and RL, and were about 21 and 9% higher than under WL, while under BL the MDA accumulation decreased by 15% (Figure 8d). 

### 3.7. Accumulation of Leaf Proteins Related to NPQ

The immunoblot analysis demonstrated that light quality changed the accumulation patterns of analyzed proteins. As shown in Figure 9, plants that were grown under monochromatic red and blue light showed increased levels of PsbS protein by about 48 and 89%, respectively, while the lowest content of PsbS was recorded in plants grown under GL (reduced by 20%) with respect to WL (Figure 9a,b, Appendix A). In addition, VDE accumulation was 34 and 50% higher in RL and BL plants than in control plants. In contrast, there was a 15% decrease in VDE content in GL-grown leaves (Figure 9a,c, Appendix A).

Additionally, no significant differences in PGRL1 concentrations were observed between leaves of RL- and BL-grown tomato plants, but the protein level was still 11% higher than in WL plants (Figure 9a,d, Appendix A). The lowest level of PGRL1 protein was again found in GL plants, with a decrease of nearly 8%. The plants grown under RL and BL showed approximately 10% higher cyt*f* accumulation than plants grown under the mixed spectrum, while plants grown under GL showed a 30% decrease in cytochrome subunit concentration (Figure 9a,e, Appendix A).

## 4. Discussion

### 4.1. Photosynthetic Capacity under Different Type of Light Quality

As compared to WL, there was a significant decrease in Fv/Fm value in RL- and GL-grown plants, while BL improved the maximum PSII photochemical efficiency. However, in all light treatments, the Fv/Fm value did not exceed the 0.8 value, which can be, at least partially, caused by increased light intensity applied during the pre-illumination of dark-adapted plants [26]. Authors of [26] documented that for plants grown at 150 µmol m^−2^ s^−1^ reduction of Fv/Fm was detectable when applied more than 300 µmol m^−2^ s^−1^, and pronounced when applied >1000 µmol m^−2^ s^−1^ of light during the pre-illumination. Thus, since we have grown plants under lower PPFD, the light intensity of 500 µmol m^−2^ s^−1^ could, in fact, reduce the maximum photochemical efficiency of PSII in all analyzed groups. However, this approach allowed us to overcome the limitation of NPQ formation by the rate of Z synthesis and activation of photosynthetic electron transport [26].

At the same time, the effective quantum yield of PSII photochemistry (ΦPSII) measured during the dark recovery was higher in WL and BL plants, while ΦNO was lower in these plants, compared to RL and GL. This might indicate a sort of photoinactivation in PSII in RL and GL plants after exposition to high-intensity light due to photoprotective mechanisms impairment [29]. However, while the ΦNPQ was indeed lower in GL, its amplitude in RL was significantly increased compared to WL. Moreover, analyzed dynamics of ΦNO in water-incubated leaves of all light treatments (Figure 7) indicate the relative stability of its value during measurement, which results from compensatory changes in ΦPSII (Figure 5) and ΦNPQ (Figure 6). This is because the sum of all yields for dissipative processes is unity [29]. Whereas, the observed discrepancies of ΦNO amplitude among the light treatments, likely arise from the concomitant competing activity of ΦNPQ positively correlated to the level of PsbS protein accumulation [29].

Therefore, to elucidate the mechanism by which monochromatic red and green light reduced the photochemical efficiency we also analyzed chlorophyll *a* and *b* and carotenoids accumulation. Analysis showed, however, that among the monochromatic lights, only green exerted a significant negative influence on pigment accumulation, while the effect of RL was not statistically significant. The highest chlorophyll accumulation was noticed for BL plants. Consequently, as the WL spectrum is composed of individual R-, G- and B-LEDs, we conclude that the observed improvement in photochemical activity and pigment accumulation in WL was mostly related to the contribution of blue light in the spectrum. Additionally, the increased concentrations of cyt*f* and PGRL1 proteins documented in monochromatic BL and RL indicate that the rates of electron transfer under BL and RL were significantly higher than in GL and even more enhanced than under mixed RGB spectrum. Previously, the authors of [17] documented that, in rice seedlings exposed to monochromatic R, G, or B light, the Fv/Fm levels were the same as in WL. However, rice grown in red or green LED showed low efficiency of photochemical utilization of absorbed light energy, as indicated by decreased ΦPSII [17]. It was also proved [41] that an increasing percentage of BL in the RB spectrum substantially increased both Fv/Fm and ΦPSII, while leaves developed under monochromatic R light presented suboptimal Fv/Fm value. 

Interestingly, we found that infiltration of leaf samples with ascorbate prior to pre-illumination increased Fv/Fm in GL and BL plants, compared to water control and improved ΦPSII recovery during relaxation in GL-grown plants. In contrast, DTT-fed leaves of WL, GL, and BL plants showed decreased Fv/Fm and restricted ΦPSII recovery. According to previous research [42], AsA plays a critical role in protecting the photosynthetic apparatus from high light damage. It has been documented [43] that ascorbate-fed leaves showed a smaller decrease and that dithiothreitol-fed leaves showed a greater decrease in Fv/Fm in response to stress conditions. Thus, it seems that feeding ascorbate to GL leaves ameliorated the PSII photodamage during the pre-illumination phase due to increased ΦNPQ along with restriction of non-regulated energy dissipation, ΦNO. This implies that GL-grown plants might suffer from restricted availability of reduced ascorbate, especially as the AsA used for leaf-feeding exist mainly in its reduced, active status [43].

### 4.2. Effect of Light Quality on qE NPQ Component

Previous analyses [17] documented that blue LED treatment increased and that red and green treatments decreased the NPQ levels compared to white LED treatment. Consistent with this, we provide evidence that growth light quality alters the kinetics and amplitude of the rapidly reversible non-photochemical quenching component, qE. The more rapid kinetics of the fast phase of NPQ induction and relaxation is related to the increased level of PsbS protein [2]. Moreover, as the active form of PsbS is triggered by a low lumen pH we also analyzed cytochrome *b*_6_*f* subunit f (cyt*f*) and the PGR5/PGRL1 complex subunit PGRL1. cyt*b*_6_*f* occupies a central position in photosynthetic electron transport in LET and CET and determines the rate of protons translocation from the stroma to the lumen via the Q-cycle [44]. PGRL1, a transmembrane protein present in thylakoids and coupled with PGR5 protein [45], mediates the main CET pathway [46], that moves H^+^ into the thylakoid lumen via the Q-cycle [10,47]. 

Most interestingly, we found that BL-developed leaves and, to a lesser extent, RL leaves, presented significantly increased levels of PsbS, cyt*f*, and PGRL1, compared to WL. At the same time, plants grown under monochromatic green light or mixed red–green–blue spectrum had significantly diminished levels of PsbS, cyt*f*, and PGRL1 protein. The importance of the PsbS and PGR5/PGRL1 complex in NPQ activation was previously documented with *npq4* and *pgr5* (also *pgr1*) mutants, which are defective in PsbS-dependent NPQ and proton influx from the stroma to the lumen [48], respectively. Several authors [7,26,47,48,49] have documented that accelerated NPQ formation induced by pre-illumination was absent in *npq4*, which presented almost no NPQ formation, while in the *pgr1* mutant, the NPQ rise was visibly slowed down and sensitive to DTT feeding [26]. On the other hand, another study [2] documented that *Arabidopsis thaliana* line *L17*, which overexpresses PsbS, presented an increased amplitude of rapidly reversible quenching and recovery. As the rapid turn-off of qE is also related to PsbS protein [50], the increased PsbS accumulation noticed in BL and RL plants may enhance the kinetics of both the onset and recovery of NPQ. 

Thus, it might be speculated that a lower amplitude and a slower rate of NPQ formation in GL and WL are predominantly due to a decreased PsbS level and a reduced rate of ΔpH formation. In contrast to previous analysis [26], however, we found that DTT infiltration was less effective in preventing rapid NPQ induction in GL plants (35%), compared to WL (67%), BL (64%), or RL (63%). Moreover, as the AsA feeding in GL leaves significantly induced qE, we speculate that the reduced rate of qE NPQ observed in GL is a consequence of the combination of reduced PsbS and decreased VDE activity rather than depressed pH gradient formation. Another study [51] also employed DTT feeding to gain insight into the role of Z in fast NPQ induction and proved that Z formation during pre-illumination is implicated in qE modulation_._ It is widely accepted that Z is required for the generation of maximum qE, whereas the amount of Z needed for such activation is rather low [26]. The authors of [26] defined such process as transient qE (qE_TR_) and noted that qE_TR_ shows similar characteristics to qE, while it depends on the presence of PsbS protein, transthylakoid pH gradient, and the amount of de-epoxidized xanthophylls. Moreover, other authors [7] proposed that Z can be considered as a kind of light stress memory in chloroplasts, allowing rapid reactivation of photoprotective NPQ processes. In our study, however, the different amplitude of the NPQ immediate response was related to growth light adaptations expressed in protein alterations, rather than the pre-illumination history. We found that monochromatic RL and, even more, the BL condition might, at some point, mimic the light stress conditions, enhancing the accumulation of NPQ-related proteins, consequently, presenting an enhanced NPQ response. An opposite trend was observed for GL- and WL-developed leaves, indicating the alleviating effect of the green component in the RGB spectrum on WL plants, despite the pre-illumination. 

### 4.3. Effect of Light Quality on qZ NPQ Component

In our study, the slower phase of NPQ induction was attributed to qZ, as the incubation of leaves with DTT successfully eliminated the NPQ rise on the 50–600 s timescale compared to water-treated samples. Additionally, as the amplitude of qZ did not increase at longer illumination times (600–1250 s), the photoinhibitory NPQ component (qI) might be excluded [27]. Overall, the qZ phase of water-treated samples contributed about 16% of the total NPQ value in BL and about 24% in GL leaves. Thus, the effect of inhibited Z formation on qZ was less pronounced than observed for fast qE induction, which has the largest impact on the quenching rate. However, for the GL leaves, we found that AsA feeding also increased the amplitude of the qZ component. Therefore, it seems that Z synthesis was not completely abolished in GL samples, but rather significantly slowed down, and such an inhibition cannot be easily overcome by pre-illumination. It was suggested in [52] that light wavelengths are not equally effective in stimulating carotenoid de-epoxidation upon pre-illumination. That study documented that green light at low intensity was ineffective in inducing Z synthesis, compared to R light. Interestingly, even though pre-illumination in our study was conducted with an RGB light source, we observed decreased NPQ dynamics in leaves developed previously under monochromatic green or green-supplemented RGB spectrum. The most reasonable hypothesis, in this context, would seem to be that GL imposes an inactivation of Z synthesis with VDE, which might be only partially rescued with AsA feeding. To solve this, we analyzed VDE accumulation and found the both GL and WL-grown leaves presented significantly lower enzyme levels compared to BL and RL. Similarly, other authors [17] have documented that rice seedlings grown under blue LEDs presented upregulated expression, and those grown under green and red LEDs presented unchanged expression of the VDE gene, compared to white LED.

Additionally, the kinetics of the middle phase of NPQ relaxation (1300–1540 s) seems to be partially attributable to the Z pool, as the rate of relaxation was substantially lower in the DTT-fed leaves than in the control condition. It has been reported [8] that DTT causes a strong dose-dependent decrease in the amplitude of NPQ dark relaxation due to its interference with the chloroplast redox state. The retardation of epoxidation kinetics in DTT samples might also be correlated with the decrease in PSII quantum efficiency [53] observed in WL, GL, and BL plants. 

### 4.4. Correlation between Antioxidant Activity, Lipid Peroxidation and Excitation Energy Quenching under Different Light Quality

To further explain the way AsA infiltration stimulated NPQ of plants grown under different types of light quality, we analyzed the activity of the antioxidant enzymes. It has been reported [1] that the ascorbate limitation of VDE and NPQ that occurs under stress conditions is affected by APX activity. As AsA acts as an electron donor for both de-epoxidation of V and neutralization of chloroplastic H_2_O_2_, APX competes with VDE for the AsA pool [54]. Second, the way that light quality influences APX and other related antioxidant enzymes (such as SOD and CAT) has been previously analyzed [16,17]. Similar to previous reports [16,17], our results also indicate that spectrum quality has a rather minor influence on CAT activity. The authors of [16] concluded, however, that including GL in the RB spectrum increased the activity of APX and SOD, which can scavenge ROS, whereas others [17] have noticed that both monochromatic red and green light increase APX and SOD activity compared to white and blue light treatment. As an explanation, the latter authors [17] stated that GL- and RL-induced APX and SOD activity was due to insufficient energy dissipation through zeaxanthin and NPQ. Interestingly, we also noticed that increased APX and SOD activities in GL and WL leaves were related to a concomitant depressed NPQ amplitude. Similarly, the authors of [5] also documented that rice grown under blue or red light presented a compromised antioxidant system compared to white light, which contributed to increased NPQ amplitude. In contrast to the results of the previous study [17], however, we did not notice a stimulating influence of RL on antioxidant enzyme activity.

Overall, our results suggest that the influence of monochromatic G, and, to a lesser extent, its addition to the RB spectrum is manifested in an increased demand for APX and SOD activity due to insufficient dissipative processes. Moreover, as we noticed increased MDA accumulation in GL- and RL-developed leaves, we speculate that both suffered from increased oxidative damage to chloroplast membranes caused by excessive ROS formation, in the way that has been previously reported [55]. These findings are not strictly concurrent with antioxidant activities pattern, explain, however, a concomitant decrease in Fv/Fm value in response to different light treatments. Previous authors [56] stated that the reduction in Fv/Fm indicates that the plant is suffering from a suboptimal light environment and that disturbed ROS homeostasis induces antioxidant components and oxidative damage. Additionally, as a reduced Fv/Fm ratio means that the PSII activity already decreased in the initial process of photoseparation of charges in its reaction center [57], it might also explain the reduced photochemistry yield (ΦPSII) value observed under monochromatic red and green light. As the authors of [57] revealed a critical role of various photoreceptors in the photosynthetic response of tomato plants to short-term light stress, an additional analysis of photoreceptors (e.g., phytochromes, cryptochromes) in plants grown under different light qualities is worthy of consideration for further study.

## 5. Conclusions

The results demonstrate that monochromatic red and blue light increased the kinetics and amplitude of NPQ compared to what was observed for GL- and WL-developed leaves. Moreover, the BL- and RL-developed leaves presented significantly increased levels of PsbS, cyt*f*, and PGRL1, while the plants grown under monochromatic GL or mixed red–green–blue spectrum had significantly diminished levels of accumulation. We also noticed that AsA feeding was able to stimulate NPQ and ΦNPQ in WL and GL leaves, which was associated with enhanced APX activity and decreased VDE activity in these plants. On the contrary, growing plants under RL or BL attenuated the APX and SOD (but not CAT) activity along with increased VDE activity. Interestingly, the effects of inhibited Z formation with DTT in pre-illuminated samples were more pronounced during rapid qE than qZ induction, indicating the importance of de-epoxidized xanthophylls during fast NPQ formation in light. Finally, we found that impairment of the photosynthetic apparatus was enhanced in plants exposed to monochromatic RL or GL, as evidenced by decreased photochemistry efficiency (Fv/Fm and ΦPSII), increased MDA accumulation, and in GL plants also by decreased photosynthetic pigment levels. While the WL and BL-grown plants presented higher vitality of PSII and a lower rate of lipid peroxidation.

## Figures and Tables

**Figure 1 biology-10-00721-f001:**
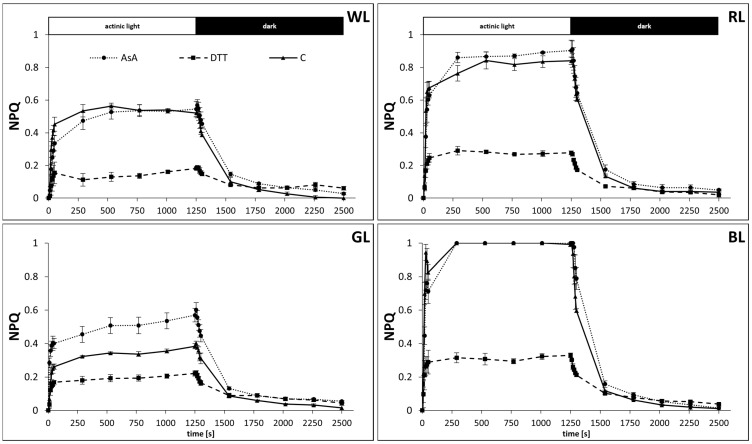
Dynamic changes of non-photochemical quenching, NPQ, in leaves of tomato plants (*Solanum lycopersicum* L. cv. Malinowy Ozarowski) grown under different LED light conditions (see Material and Methods for details). 10 mm diameter leaf discs were infiltrated with 10 mM ascorbic acid (AsA) or 5 mM dithiothreitol (DTT), or distilled water (C, control) for 30 min in the dark, followed by pre-illumination at 500 µmol m^–2^ s^–1^ of RGB light for next 30 min. The dynamics of the NPQ were determined in re-darkened leaf samples illuminated for 20 min at 185 µmol m^–2^ s^–1^ of blue actinic light (AL = 450 nm) (induction), followed by 20 min of dark incubation (relaxation), and derived from the maximum fluorescence induced by saturation pulses (5000 µmol m^−2^ s^−1^, 0.8 s) at each of the measuring points. Each data point represents the average ± SD of six independent measurements (*n* = 6).

**Figure 2 biology-10-00721-f002:**
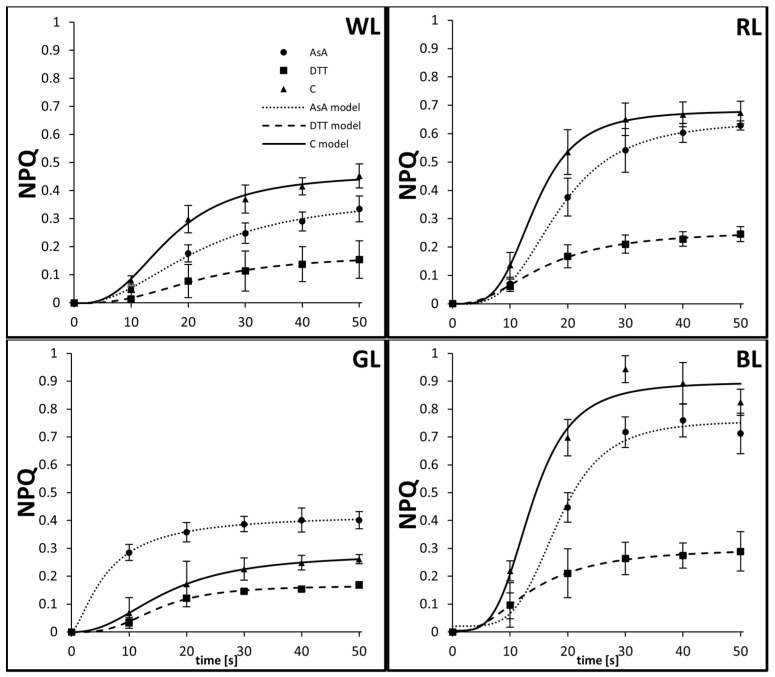
Dynamic changes of fast induction of non-photochemical quenching, NPQ, in leaves of tomato plants (*Solanum lycopersicum* L. cv. Malinowy Ozarowski) grown under different LED light conditions (see Material and Methods for details). 10 mm diameter leaf discs were infiltrated with 10 mM ascorbic acid (AsA) or 5 mM dithiothreitol (DTT), or distilled water (C, control) for 30 min in the dark, followed by pre-illumination at 500 µmol m^–2^ s^–1^ of RGB light for next 30 min. The dynamics of the NPQ induction were determined in re-darkened leaf samples and derived from saturation pulses (5000 µmol m^–2^ s^–1^, 0.8 s) applied every 10 s during the first 50 s of blue actinic light (AL = 450 nm) illumination. Each data point represents the average ± SD of six independent measurements (*n* = 6). The logistic regression model was employed to fit the experimental data of fast (50 s) qE NPQ induction in the AsA-, DTT- or water-treated samples. Fitting was applied as specified in Material and Methods and reported with an adjusted R^2^ (R_a_^2^) value (Table 1) to determine the goodness of data fitting.

**Figure 3 biology-10-00721-f003:**
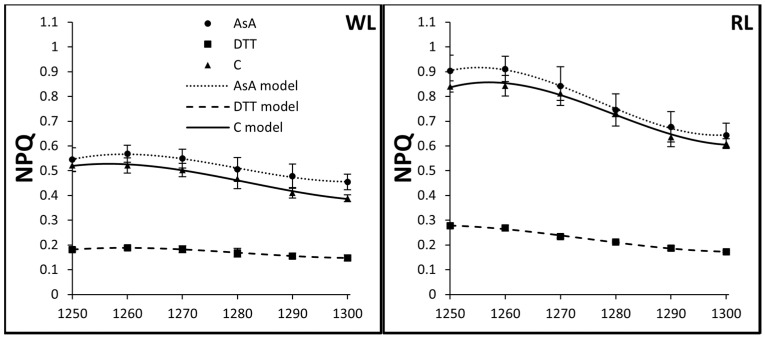
Dynamic changes of fast relaxation of non-photochemical quenching, NPQ, in leaves of tomato plants (*Solanum lycopersicum* L. cv. Malinowy Ozarowski) grown under different LED light conditions (see Material and Methods for details). 10 mm diameter leaf discs were infiltrated with 10 mM ascorbic acid (AsA) or 5 mM dithiothreitol (DTT), or distilled water (C, control) for 30 min in the dark, followed by pre-illumination at 500 µmol m^−2^ s^−1^ of RGB light for next 30 min. The dynamics of the NPQ relaxation were determined in leaf samples immediately after actinic light turn off and derived from saturation pulses (5000 µmol m^−2^ s^−1^, 0.8 s) applied every 10 s during the first 50 s of darkness. Each data point represents the average ± SD of six independent measurements (*n* = 6). The cubic polynomial regression model was employed to fit the experimental data of fast (50 s) qE NPQ relaxation in the AsA-, DTT- or water-treated samples. Fitting was applied as specified in Material and Methods and reported with an adjusted R^2^ (R_a_^2^) value (Table 2) to determine the goodness of data fitting.

**Figure 4 biology-10-00721-f004:**
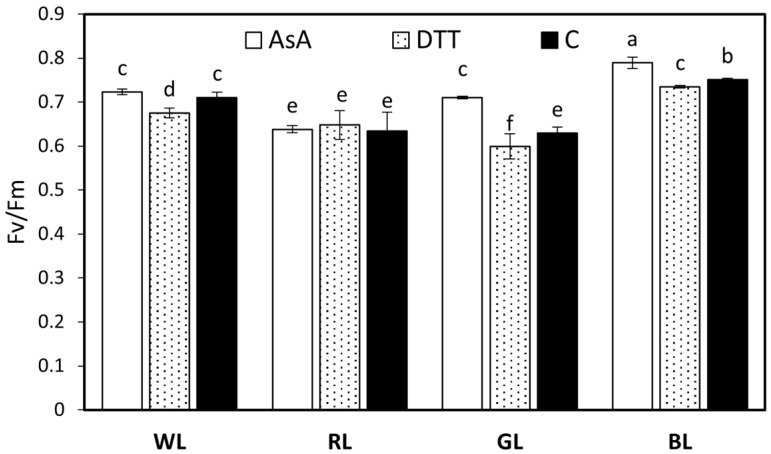
The maximal photochemical yield of PSII, Fv/Fm, in leaves of tomato plants (*Solanum lycopersicum* L. cv. Malinowy Ozarowski) grown under different light conditions (see Material and Methods for details). 10 mm diameter leaf discs were infiltrated with 10 mM ascorbic acid (AsA) or 5 mM dithiothreitol (DTT), or distilled water (C, control) for 30 min in the dark, followed by pre-illumination at 500 µmol m^−2^ s^−1^ of RGB light for next 30 min. The minimal fluorescence level (Fo) was measured by the measuring modulated blue light (450 nm), which was sufficiently low (0.01 µmol m^−2^ s^−1^) to not induce any significant variable fluorescence. The maximal fluorescence level (Fm) was determined by a 0.8 s saturating blue light pulse (SP = 450 nm) at 5000 µmol m^−2^ s^−1^ in dark-adapted samples. The maximum PSII photochemical efficiency Fv/Fm was derived from that (Fv/Fm = (Fm − Fo)/Fm). Each bar represents the average ± SD of six independent measurements (*n* = 6). Different letters indicate significant differences between treatments (*p* = 0.05) with a Tukey’s HSD.

**Figure 5 biology-10-00721-f005:**
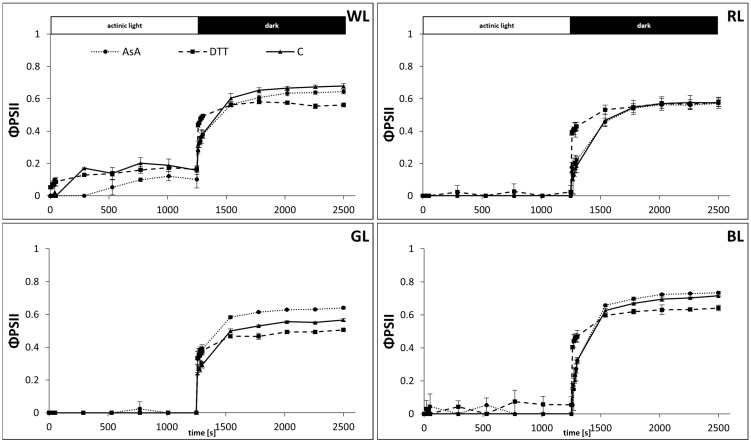
Measurement of the effective quantum yield of PSII, ΦPSII, in leaves of tomato plants (*Solanum lycopersicum* L. cv. Malinowy Ozarowski) grown under different light conditions (see Material and Methods for details). 10 mm diameter leaf discs were infiltrated with 10 mM ascorbic acid (AsA) or 5 mM dithiothreitol (DTT), or distilled water (C, control) for 30 min in the dark, followed by pre-illumination at 500 µmol m^−2^ s^−1^ of RGB light for next 30 min. The dynamics of the ΦPSII were determined in re-darkened leaf samples illuminated for 20 min at 185 µmol m^−2^ s^−1^ of blue actinic light (AL = 450 nm) (induction), followed by 20 min of dark incubation (relaxation), and derived from the maximum fluorescence induced by saturation pulses (5000 µmol m^−2^ s^−1^, 0.8 s) at each of the measuring points (ΦPSII = (Fm′ − F)/Fm′). Each data point represents the average ± SD of six independent measurements (*n* = 6).

**Figure 6 biology-10-00721-f006:**
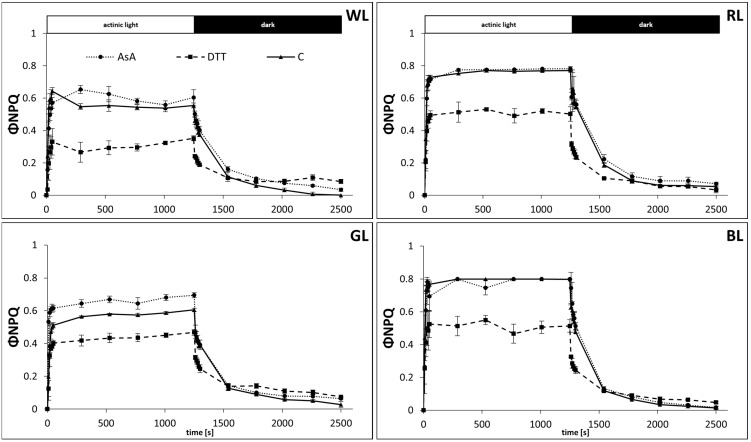
Measurement of the quantum yields of regulated energy dissipation, ΦNPQ, in leaves of tomato plants (*Solanum lycopersicum* L. cv. Malinowy Ozarowski) grown under different light conditions (see Material and Methods for details). 10 mm diameter leaf discs were infiltrated with 10 mM ascorbic acid (AsA) or 5 mM dithiothreitol (DTT), or distilled water (C, control) for 30 min in the dark, followed by pre-illumination at 500 µmol m^−2^ s^−1^ of RGB light for next 30 min. The dynamics of the ΦNPQ were determined in re-darkened leaf samples illuminated for 20 min at 185 µmol m^−2^ s^−1^ of blue actinic light (AL = 450 nm) (induction), followed by 20 min of dark incubation (relaxation), and derived from the maximum fluorescence induced by saturation pulses (5000 µmol m^−2^ s^−1^, 0.8 s) at each of the measuring points (ΦNPQ = 1 − ΦPSII − ΦNO). Each data point represents the average ± SD of six independent measurements (*n* = 6).

**Figure 7 biology-10-00721-f007:**
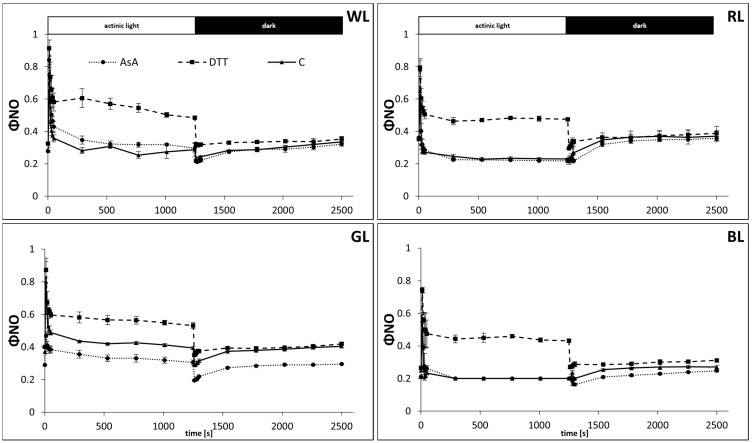
Measurement of the quantum yields of non-regulated energy dissipation in PSII, ΦNO, in leaves of tomato plants (*Solanum lycopersicum* L. cv. Malinowy Ozarowski) grown under different light conditions (see Material and Methods for details). 10 mm diameter leaf discs were infiltrated with 10 mM ascorbic acid (AsA) or 5 mM dithiothreitol (DTT), or distilled water (C, control) for 30 min in the dark, followed by pre-illumination at 500 µmol m^−2^ s^−1^ of RGB light for next 30 min. The dynamics of the ΦNO were determined in re-darkened leaf samples illuminated for 20 min at 185 µmol m^−2^ s^−1^ of blue actinic light (AL = 450 nm) (induction), followed by 20 min of dark incubation (relaxation), and derived from the maximum fluorescence induced by saturation pulses (5000 µmol m^−2^ s^−1^, 0.8 s) at each of the measuring points [ΦNO = 1/(NPQ + 1 + qL(Fm/Fo − 1))]. Each data point represents the average ± SD of six independent measurements (*n* = 6).

**Figure 8 biology-10-00721-f008:**
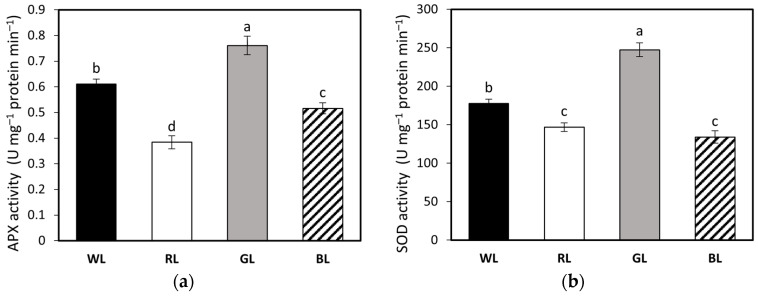
Ascorbate peroxidase (APX, **a**), superoxide dismutase (SOD, **b**) and catalase (CAT, **c**) enzyme activity levels, and malondialdehyde (MDA, **d**) accumulation in leaves of tomato plants (*Solanum lycopersicum* L. cv. Malinowy Ozarowski) grown under different light conditions (see Material and Methods for details). Bars show the means ± SD (*n* = 4). Different letters indicate significant differences between treatments (*p* = 0.05) with a Tukey’s HSD test. U—unit of enzyme activity. FW—fresh weight.

**Figure 9 biology-10-00721-f009:**
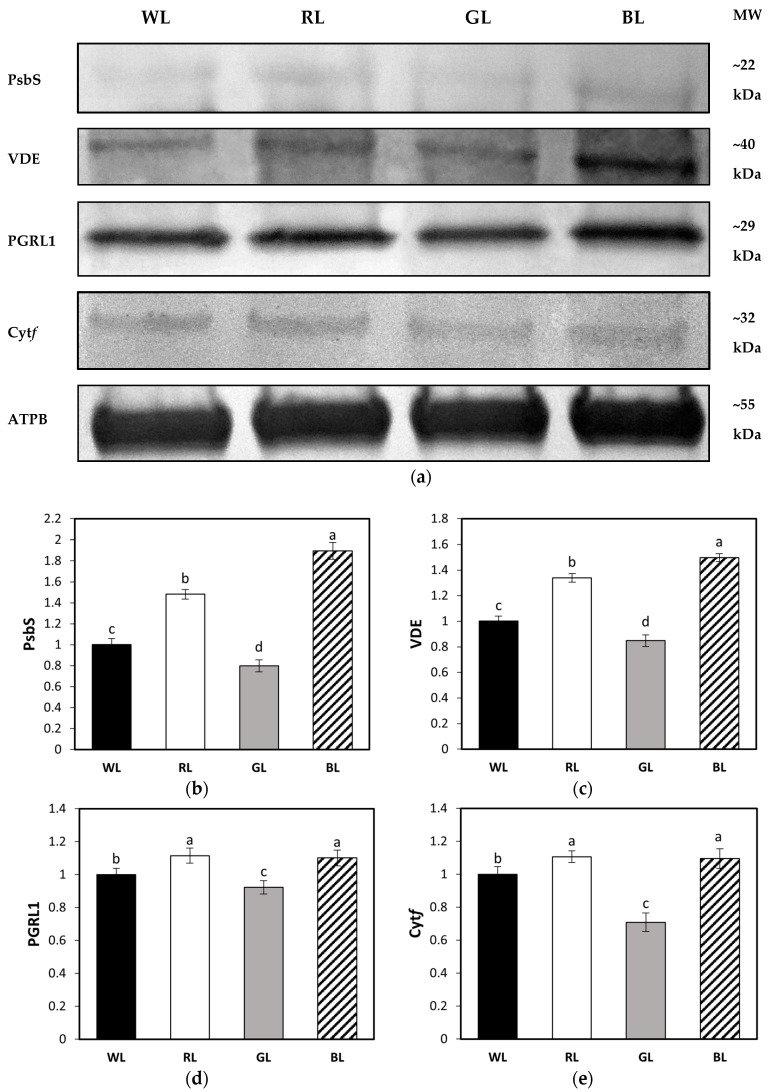
Western Blot analyses (**a**) and densitometric analyses of PsbS (**b**), VDE (**c**), PGRL1 (**d**), and cyt*f* (**e**) proteins in leaves of tomato plants (*Solanum lycopersicum* L. cv. Malinowy Ozarowski) grown under different light conditions (see Material and Methods for details). The bands were normalized to the appropriate β subunit of ATP synthase (ATPB) band (loading control, Appendix A) (**a**). The bar diagrams (**b**–**e**) represent pixel volumes (densitometric analyses) of proteins in samples. Each value represents the mean ± SD (*n* = 3) considering the control sample (WL) value as 1 (100%). Different letters indicate significant differences between treatments (*p* = 0.05) with a Tukey’s HSD test. MW—molecular weight.

**Table 1 biology-10-00721-t001:** The parameters of the logistic regression model employed to fit the experimental data (*x*) of fast (50 s) qE NPQ induction in the AsA-, DTT- or water-treated samples (C). The fit equation: $y=\frac{A_{1}-A_{2}}{1+{\left(\frac{x}{x_{0}}\right)}^{\alpha }}+A_{2}$, where *A*_1_—minimum asymptote, *A*_2_—maximum asymptote, *x*_0_—the value of the inflection point, *α*—a slope of the logistic growth rate (steepness of the curve), R_a_^2^—the adjusted r-square value, determining the goodness of fit, accounts for the degrees of freedom. WL—the R:G:B = 1:1:1 at 120 µmol m^–2^ s^–1^; RL, GL or BL—monochromatic red (R), green (G) or blue (B) light (L) at 80 µmol m^–2^ s^–1^, respectively.

Light Quality	Treatment	*A_1_*	*A_2_*	*x_0_*	*α*	R_a_^2^
WL	AsA	−0.00252	0.38013	22.09031	2.22938	0.99203
	DTT	−0.00167	0.17035	22.17636	2.58926	0.99382
	C	−0.00274	0.46039	16.73017	2.78761	0.99076
RL	AsA	−0.00147	0.64659	18.21531	3.37634	0.99971
	DTT	−3.29693 × 10^−4^	0.25832	15.6885	2.34694	0.99808
	C	3.59786 × 10^−4^	0.68169	14.29079	3.89071	0.99978
GL	AsA	1.23493 × 10^−5^	0.42252	6.1032	1.47486	0.99937
	DTT	−4.9053 × 10^−4^	0.16698	14.89135	3.05313	0.99232
	C	6.50152 × 10^−6^	0.2812	16.37096	2.30662	0.99995
BL	AsA	0.02089	0.75903	18.24241	4.49567	0.9808
	DTT	9.39587 × 10^−5^	0.30166	13.81813	2.34655	0.99839
	C	0.00485	0.8969	13.65348	3.89107	0.95673

**Table 2 biology-10-00721-t002:** The parameters of the polynomial cubic regression model employed to fit the experimental data (*x*) of fast (50 s) qE NPQ relaxation in the AsA-, DTT- or water-treated samples (C). The fit equation: $y=A+Bx+Cx^{2}+Dx^{3}$, where *A*—offset, *B*, *C* and *D*—coefficients, R_a_^2^—the adjusted r-square value, determining the goodness of fit, accounts for the degrees of freedom. WL—the R:G:B = 1:1:1 at 120 µmol m^–2^ s^–1^; RL, GL or BL—monochromatic red (R), green (G) or blue (B) light (L) at 80 µmol m^–2^ s^–1^, respectively.

Light Quality	Treatment	*A*	*B*	*C*	*D*	R_a_^2^
WL	AsA	−6808.2723	15.96759	−0.01248	3.25 × 10^−6^	0.98669
	DTT	−2209.638	5.17863	−0.00404	1.05247 × 10^−6^	0.99798
	C	−4908.0904	11.49698	−0.00897	2.33333 × 10^−6^	0.97961
RL	AsA	−13877.6622	32.62703	−0.02556	6.67284 × 10^−6^	0.99688
	DTT	−2583.3524	6.08759	−0.00478	1.25 × 10^−6^	0.98621
	C	−11791.3213	27.66722	−0.02163	5.63579 × 10^−6^	0.98203
GL	AsA	−8712.6496	20.45127	−0.016	4.16975 × 10^−6^	0.96758
	DTT	−2915.1715	6.84801	−0.00536	1.39815 × 10^−6^	0.99193
	C	−5768.1806	13.53744	−0.01059	2.75926 × 10^−6^	0.95141
BL	AsA	−1902.5802	4.30528	−0.00324	8.0864 × 10^−7^	0.90916
	DTT	−620.6656	1.50338	−0.00121	3.24074 × 10^−7^	0.98593
	C	−14325.4182	33.57812	−0.02622	6.82407 × 10^−6^	0.99645

**Table 3 biology-10-00721-t003:** The abundance of photosynthetic pigments extracted with DMSO from 3-mm leaf discs of tomato plants grown under different light conditions: WL—the R:G:B = 1:1:1 at 120 µmol m^−2^ s^−1^; RL, GL or BL—monochromatic red (R), green (G) or blue (B) light (L) at 80 µmol m^−2^ s^−1^, respectively. The presented values are means of ten replicates ± SD. Different letters in the same row indicate significant differences between treatments at *p* = 0.05 with a Tukey’s HSD test. Chl *a*—chlorophyll *a*, Chl *b*—chlorophyll *b*, FW—fresh weight.

Parameter	Treatment			
	WL	RL	GL	BL
Chl *a* + *b* (mg g^−1^ FW)	2.77 ± 0.19 ^a^	2.62 ± 0.23 ^ab^	2.42 ± 0.13 ^b^	2.85 ± 0.18 ^a^
Chl *a* (mg g^−1^ FW)	2.18 ± 0.20 ^ab^	2.05 ± 0.19 ^ab^	1.91 ± 0.16 ^b^	2.27 ± 0.19 ^a^
Chl *b* (mg g^−1^ FW)	0.59 ± 0.02 ^a^	0.57 ± 0.03 ^a^	0.51 ± 0.03 ^b^	0.58 ± 0.03 ^a^
Chl *a*/*b*	3.69 ± 0.44 ^a^	3.60 ± 0.37 ^a^	3.75 ± 0.48 ^a^	3.91 ± 0.46 ^a^
Carotenoids (mg g^−1^ FW)	0.44 ± 0.03 ^a^	0.42 ± 0.03 ^a^	0.37 ± 0.02 ^b^	0.43 ± 0.03 ^a^

## Data Availability

Data are available on request due to restrictions, e.g., privacy or ethics. The data presented in this study are available on request from the corresponding author. The data are not publicly available due to the strict management of various data and technical resources within the research teams.

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
