# Peer review of "Light Quality-Dependent Regulation of Non-Photochemical Quenching in Tomato Plants"

_biology, 2021, doi:10.3390/biology10080721_

Round 1
Reviewer 1 Report
The manuscript by Trojak and Skowron describes the effect of different spectral composition of growing light (blue, green, red, and white light) on a regulation of photosynthetic function (mainly non-photochemical quenching) in tomato plants, which is important to consider when LED illumination is used for a plant cultivation.
The results are clearly presented and discussed with a relevant literature. To give the obtained results a greater physiological relevance and to better describe the impact of different spectral composition of growth light on a photosynthetic performance in tomato plants, additional parameters could be included in the revised version of the manuscript.
Firstly, the authors observed a positive effect of blue light e.g. on a photochemical function of photosystem II compared to green or red light. Was there any visible positive effect of blue light e.g. on a plant growth (a biomass production/fresh weight)?
Secondly, the authors concluded: “Finally, we found a noticeable disruption of photosynthetic apparatus in plants exposed to monochromatic red light or green light, noted with decreased photochemistry efficiency, whereas in white light and blue light-grown plants the vitality of PSII was enhanced.” Can authors better specify the disruption of photosynthetic apparatus? At least analysis of a pigment content in different plants could provide important information about the composition of photosynthetic apparatus.
Reviewer 2 Report
Comments to the MS of Magdalena Trojak and Ernest Skowron: “ Light Quality-Dependent Regulation of Non-photochemical Quenching in Tomato Plants”
The present study investigated the impact of light spectral quality on the dynamics of non-photochemical quenching, photosynthetic parameters, and the absorbed light energy partitioning in PSII introducing into tomato leaves DTT and AsA. Also, in the MS content of PsbS, VDE, PGRL1 and cytf proteins in tomato plants leaves associated with system of dissipation of excess energy was evaluated. I appreciate the data, which are important for understanding of mechanism of quenching in plants. However, in my opinion, it would be good to explain in more detail, why DTT and AsA were used in the experiments since these substances have many functions and possible effects.
In addition, why values of Fv/Fm in tomato leaves were so small in all variants compared to known values of (from literature) the yields in tomato plants (0.82, for instance, Kreslavski et al. 2020, Journal of Photochemistry and Photobiology B: Biology)
FPSII in all variants were also small; especially, they were small in green and red treatments. Why?
Why the authors did determine activity of antioxidant enzymes such as APX and SOD. Many works determine activity of POD and catalase, as example. Please, justify it.
What can you say on stress conditions. The quantum yield in white light plants is 0.70, which characterizes the state of PA as stress. The grade of the stress depends on the light intensity that can be different depending on the light quality. Authors have to take this into account.
Why did you use for tomato plants such light intensities: 80 mkmol photons m-2 s-1 and 120 mkmol photons m-2 s-1?.
Lines 683-686. “Finally, we found a noticeable disruption of photosynthetic apparatus in plants exposed to monochromatic RL or GL, noted with decreased photochemistry efficiency, whereas in WL and BL-grown plants the vitality of PSII was enhanced”. The statement is unclear since in all variants, the value of Fv/Fm is within 0.62-0.74, that means a damage to photosynthetic apparatus but the grade of the damage is various.
In addition, please describe a method of evaluation of stress at the level of photosynthetic apparatus.
Please. Discuss the data based on the formulae that FPSII+F(NO)+F(NPQ) =1 (Kramer et al. 2004).
Minor comments
Lines 402-403. It is written: “the value of Fv/Fm is highly consistent, with values of about 0.83, and correlates to the maximum quantum yield of photosynthesis. I advise to write: “the value of Fv/Fm photosynthesis.
Lines 32-34. The sentence is not quite clear: “Results confirmed that monochromatic BL increased the quantum efficiency of PSII, while the RL and BL significantly enhanced the NPQ amplitude and showed negative effects on antioxidant enzyme activities.
The MS looks well with point of view methods and results obtained in the MS and theoretical considerations of mechanisms of formation of NPQ. However, the main lack is weak physiological description of approach and results obtained in the MS. I advise to improve that using my comments.
Reviewer 3 Report
This manuscript subject matter fulfills the general scope of the journal and its also original research. In this manuscript entitled " Light Quality-Dependent Regulation of Non-photochemical Quenching in Tomato Plants” provide the details on the investigation of monochromatic LEDs—red, green, and blue ones on the photoprotective and photosynthetic properties of tomato plants. The research findings highlighted the the impact of light quality on the mechanisms of regulated energy quenching and accumulation of NPQ-related proteins—photosystem II (PSII) subunit S (PsbS), PROTON GRADIENT REGULATION-LIKE1 (PGRL1), cytochrome b6f subunit f (cytf) and violaxanthin deepoxidase (VDE) which has a great prospectus to the future agriculture. I think it’s a promising strategy to cultivate indoor vegetables. In general, it is my opinion that this work can be accepted for publication in Biology after solving the following raised questions-
-The authors should carefully review the typography and correct the grammatical errors in the whole manuscript.
Title: I think title is OK.
Simple Summary: OK.
Abstract: Author should rewrite the abstract initially with properly addressing research gap, and objectives of the present research within one or two sentences.
The study indicates the impact of light quality on the mechanisms of regulated energy quenching and accumulation of NPQ-related proteins—photosystem II (PSII) subunit S (PsbS), PROTON GRADIENT REGULATION-LIKE1 (PGRL1), cytochrome b6f subunit f (cytf) and violaxanthin deepoxidase (VDE)- this sentence is meaningless and author should rewrite this sentence what actually want to address.
Keywords: The keywords are appropriate.
Introduction
The author should fully elaborated all the acronyms like PsbS, ΔpH and so on.
The author needs to include more statements about photosynthesis, light and ROS production as well mitigation strategy. The author can read the following article (https://doi.org/10.1038/s41438-020-00429-3) to address this issue.
The author wrote the introduction in a descriptive format and there is no link up in the whole introduction. There should be needs a link up among the whole introduction.
Materials and Methods
Line 112 Analysed tomato cultivar is potato leaf phenotype showing reduced leaf dissection-why author did this?
What is the basis of 10 mM ascorbic acid (AsA), and 5 mM dithiothreitol (DTT) concentration? Please add an appropriate citation.
Results
The results section is well written. If possible increase the configuration of the figure specifically figure 9.
Why author have not quantified ascorbic acid (AsA), and dithiothreitol (DTT) content in the different light spectrums as well as with ascorbic acid (AsA), and dithiothreitol (DTT) treatments plants.
Discussion
The discussion is well organized.
Conclusions
The conclusion is too long. It should be concise.
References
References are properly formatted.
Round 2
Reviewer 1 Report
The authors sufficiently addressed my remarks in the revised version.
Reviewer 2 Report
The second comments to the MS of M. Trojak and E. Skowron: “Light Quality-Dependent Regulation of Non-photochemical Quenching in Tomato Plants”
All my comments are, in whole, answered. However, I have small note.
- My comment was ¨In addition, why values of Fv/Fm in tomato leaves were so small in all variants compared to known values of (from literature) the yields in tomato plants (0.82, for instance, Kreslavski et al. 2020, Journal of Photochemistry and Photobiology B: Biology)
Authors write in response to the comments: “It should be noted, however, that the observed values of Fv/Fm, in all the treatments, have been lower than expected, presumably due to relatively short redarkening of samples prior to the measurements of chlorophyll fluorescence.”
I disagree. I think that this unfortunate explanation since even use of shorter dark-period leads often to Fv/Fm value in tomato plants, which is equal to approximately 0.80. I think that a reason of lowered value of the Fv/Fm is stressful conditions but no small dark period.
The MS can be accepted after small revision.
Reviewer 3 Report
The author addressed all the raised questions very efficiently. In this condition, the manuscript is acceptable for publication.
